# Temporal regulation of MDA5 inactivation by Caspase-3 dependent cleavage of 14-3-3η

**Yun-Jui Chan[1], Nien-Tzu Liu[1], Fu Hsin[1], Jia-Ying Lu[2], Jing-Yi Lin[2], Helene Minyi Liu[1] ***

**1** Institute of Biochemistry and Molecular Biology, College of Medicine, National Taiwan University, Taipei City, Taiwan, **2** Department of Clinical Laboratory Sciences and Medical Biotechnology, College of Medicine, National Taiwan University, Taipei City, Taiwan

* mliu@ntu.edu.tw

## Abstract

The kinetics of type I interferon (IFN) induction versus the virus replication compete, and the result of the competition determines the outcome of the infection. Chaperone proteins that involved in promoting the activation kinetics of PRRs rapidly trigger antiviral innate immunity. We have previously shown that prior to the interaction with MAVS to induce type I IFN, 14-3-3η facilitates the oligomerization and intracellular redistribution of activated MDA5. Here we report that the cleavage of 14-3-3η upon MDA5 activation, and we identified Caspase-3 activated by MDA5-dependent signaling was essential to produce sub-14-3-3η lacking the C-terminal helix (αI) and tail. The cleaved form of 14-3-3η (sub-14-3-3η) could strongly interact with MDA5 but could not support MDA5-dependent type I IFN induction, indicating the opposite functions between the full-length 14-3-3η and sub-14-3-3η. During human coronavirus or enterovirus infections, the accumulation of sub-14-3-3η was observed along with the activation of Caspase-3, suggesting that RNA viruses may antagonize 14-3-3η by promoting the formation of sub-14-3-3η to impair antiviral innate immunity. In conclusion, sub-14-3-3η, which could not promote MDA5 activation, may serve as a negative feedback to return to homeostasis to prevent excessive type I IFN production and unnecessary inflammation.

**Data Availability Statement:** All relevant data are within the manuscript and its Supporting Information files.

**Funding:** National Sciences and Technology Councils :113-2918-I-002-017 (HML) and 110-

## Summary

The protein 14-3-3η has been identified as a critical regulator in the host defense against RNA virus infections via MDA5. However, in order to restore homeostasis, the host innate immune response, particularly the production of type I interferon (IFN), needs to be turned off. The cleavage of 14-3-3η by Caspase-3 is one of negative regulatory event to downregulate type I IFN expression. Also, viruses such as coronaviruses and enteroviruses have evolved strategies to exploit 14-3-3η, impairing the immune response.

In this study, we elucidated the intricate interplay between the chaperon protein 14-3-3η, the cytosolic RNA sensor MDA5, and the protease Caspase-3. Upon virus infection, MDA5 activation triggers Caspase-3 activity, leading to the cleavage of 14-3-3η and the formation of a truncated version termed sub-14-3-3η. While sub-14-3-3η retains its interaction with MDA5, it fails to induce the production of interferon, a critical

2314-B-002-252-MY3 (HML); National Health Research Institutes: NHRI-EX111-10915SI (HML) and NHRI-EX113-11235SI (HML); National Taiwan University: 109L7810 (HML) 113L7851 (HML). The funders had no role in study design, data collection and analysis, decision to publish, or preparation of the manuscript.

**Competing interests:** The authors have declared that no competing interests exist.

component of antiviral immunity. During RNA virus infections, sub-14-3-3η are accumulated in both Caspase-3 dependent and independent manners. This discovery sheds light on how viruses manipulate host proteins to evade immune surveillance, potentially offering new targets for therapeutic intervention.

## Introduction

An early onset of type I interferon (IFN) induction and response are the keys for successful viral clearance. During viral infection, several cytosolic sensors could detect viral nucleic acids as the pathogen-associated molecular patterns (PAMPs) [1,2]. The RIG-I-like receptors (RLRs), including retinoic acid-inducible gene I (RIG-I) and melanoma differentiation-associated gene 5 (MDA5), are key cytosolic sensors to recognize PAMP RNA species [1, 2]. Once bound to non-self RNA, the RLRs undergo major conformational changes to interact with accessory proteins and downstream signaling molecules [1,2]. These protein complexes could then engage host protective innate immunity against viral infection via mitochondrial antiviral signaling protein (MAVS), a mitochondria-associated transmembrane protein, through their Caspase activation and recruitment domains (CARDs) [3,4]. We have previously showed that upon activation, RIG-I and MDA5 relocalize to mitochondria-associated membrane (MAM) to interact with MAVS, which is the crucial onset of MAVS-mediated IFN production pathway during acute phase of viral infection [3,5–7].

MDA5-mediated antiviral signaling has been shown important in the clearance of Flavivirus, Picornavirus, Paramyxovirus, and Reovirus infections [8]. In the laboratories, enterovirus infections, human coronavirus 229E infections and/or high molecular weight poly (I:C) (synthetic dsRNA) transfection are the most common agents to model MDA5 activation. It has been shown that MDA5 has an essential and nonredundant role in detecting RNA virus infection [9]. The binding of MDA5 to the long dsRNA will then cooperatively form tandem MDA5 filaments along the dsRNA, and the CARDs of MDA5 will then oligomerize to interact with MAVS for downstream antiviral signaling pathway [10]. These reports indicated that similar regulatory event of MDA5 redistribution from the cytosol to the MAM compartment promoted by 14-3-3η to interact with MAVS is critical for MDA5-mediated type I IFN induction. Through interaction with MAVS, MDA5 activation will then lead to the activation of PAMP-driven transcription factors, IFN production, and interferon-stimulated gene (ISG) expressions, resulting in the immediate onset of host antiviral state [11]. These studies indicated that not only the PRRs, signaling molecules, and transcription factors are critical, but also the chaperone proteins which facilitate the intracellular redistribution and protein-protein interactions among the signaling molecules and therefore promote the kinetics of type I IFN induction are important to restrict viral propagation and infection.

Antiviral innate immunity constitutes pro-inflammatory response to initiate adaptive immune response required for virus clearance. Yet, excessive inflammatory responses are highly destructive and could lead to tissue damage. Hence, the antiviral innate immunity requires a rapid, however, tightly regulated response to effectively clear the invading pathogens without harming the host. Several negative regulators of the type I IFN induction pathway have been reported recently, such as apoptotic Caspases and plasminogen activator inhibitor type 2 (PAI-2) (reviewed in [12]). Several studies discovered that the expression of PAI-2 was up-regulated when macrophages were stimulated by LPS [13,14], and this phenotype was dependent on IKKβ-NF-κB signaling pathway [14]. Recently, a novel role of PAI-2 in the

macrophages has been reported to suppress IL1β processing via inhibiting Caspase-1 activation in macrophages, suggesting that intracellular proteases may be involved in promoting inflammation and innate immunity [15]. Mitochondrial apoptotic pathway is known to be induced after RLR stimulations [16]. When apoptosis is triggered, the 35-kDa pro-Caspase-3 is cleaved into two active subunits of which molecular weight are 17 kDa and 12 kDa [17]. Previous studies had shown that apoptotic Caspases not only mediate proteolysis during apoptosis but also cleave the components of type I IFN induction pathway. It has been well-characterized that activated Caspase 3/7 could cleave several signaling molecules, including cGAS, MAVS, STING and IRF3, to downregulate type I IFN expression [18–20]. Indeed, the activation of Caspase-3 and other apoptotic Caspases could not only be triggered by proapoptotic signals but also be promoted by the RLR-MAVS-mediated antiviral signaling [16,21], suggesting a role of Caspase-3 as a regulator in the negative feedback loop of type I IFN induction. Pathogenic viruses, such as enteroviruses, utilize this intrinsic negative regulatory mechanism to impair type I IFN induction by promoting the activation of Caspase-3 [22]. Not only during enterovirus infections, the ectopic expression of enterovirus proteases 2A and 3C may also achieve the activation of Caspase-3 and thus downregulate the antiviral innate immunity [22,23].

We previously identified that the redistribution of RIG-I from the cytosol to a membrane fraction upon ligand recognition was controlled by 14-3-3ε, and 14-3-3η can facilitate the activation of MDA5 [5,7]. There are 7 isoforms in the 14-3-3 family, including 14-3-3β, 14-3-3γ, 14-3-3ε, 14-3-3η, 14-3-3σ, 14-3-3θ, and 14-3-3ζ. Although the cargo preferences vary among different isoforms, the tertiary structures of the 14-3-3 isoforms are well-conserved [24]. According to the structural studies of 14-3-3 proteins, a single 14-3-3 molecule is composed of nine α-helices, indicated as αA to αI [24]. The flexible loop structure between αC-αD and αH-αI are the CD loop and HI loop, respectively [24,25]. When 14-3-3 protein dimer is formed, the first four α-helices (αA to αD) are pivotal for dimerization, and the αC, αE, αG, and αI contribute to construct the target protein binding groove [26]. The ligand binding groove is a hydrophobic and positively charged patch composed by the arginine-arginine-tyrosine triad [26,27]. It was proposed that the C-terminal tail of 14-3-3, which varies in peptide lengths and sequences among all isoforms, could be the regulatory domain for the target protein binding affinity [28,29]. Because the C-terminal truncation may either enhance or impair the binding affinity of 14-3-3 ligands among different 14-3-3 isoforms, the C-terminal region as well as the αI play a critical role to regulate 14-3-3 interaction to the target proteins. Many viral proteins are also reported to interact with the 14-3-3 family. Studies have showed that both Dengue virus (DENV) and Zika virus (ZIKV) NS3 proteins use a protease-independent phosphomimetic-based mechanism to occupy 14-3-3ε and/or 14-3-3η to retain RIG-I and MDA5 in cytosol and lead to the attenuation of IFNβ induction in the infected cells [5,30], suggesting that 14-3-3 family serves as an important regulator in the type I IFN expression. It is intriguing whether other viruses may also target 14-3-3ε and/or 14-3-3η to downregulate type I IFN induction pathway and antiviral innate immunity.

In this study, we reported that MDA5 activation-dependent Caspase-3 activity cleaved 14-3-3η at the C-terminus to truncate out the αI helix of 14-3-3η. The cleaved 14-3-3η, termed as sub-14-3-3η, could strongly interact with MDA5 but could not promote MDA5-MAVS-dependent type I IFN induction. Therefore, the cleavage of 14-3-3η may serve as a negative feedback mechanism to prevent excessive type I IFN induction by MDA5-dependent signaling, and pathogenic viruses, such as human coronavirus 229E, have taken the advantage of this intrinsic pathway to downregulate MDA5-dependent antiviral signaling. Besides the antiviral innate immunity, 14-3-3η and MDA5 are both found to be associated with Rheumatic arthritis, and the serum levels of 14-3-3η can serve as the biomarker of this autoimmune disease

[31]. Understandings in the protein interaction at the molecular levels may benefit us not only in the development of antiviral drugs but also in the development of immune modulatory therapies.

## Result

### The degradation of full-length 14-3-3η and the production of sub-14-3-3η upon MDA5 activation

Our previous reports show that 14-3-3η interacts with and facilitates the signaling of full-length and the N-terminal CARDs of MDA5 (N-MDA5), which is reported as a constructively active mutant of MDA5 [5]. In our previous study, while performing anti-Flag co-immunoprecipitation (co-IP) of Myc-14-3-3η with Flag-MDA5 or Flag-N-MDA5 followed by SDS-PAGE and immunoblotting, we observed a distinct band of 14-3-3η with a faster electromobility compared to the full-length 14-3-3η by immunoblotting [5]. This observation was confirmed by the co-immunoprecipitation (co-IP) of Flag-tagged 14-3-3η and Myc-tagged MDA5 1–889, which can contribute a similar phenotype of N-MDA5 as the constructively active mutant of MDA5 (S1A and S1B Fig). Consistent to the anti-Flag co-IP results by Myc-14-3-3η and Flag-N-MDA5, Myc-MDA5 1–889 could also co-immunoprecipitate a subset of Flag-14-3-3η with faster electromobility compared to the full-length Flag-14-3-3η (S1B Fig). We then termed this distinct band of 14-3-3η as sub-14-3-3η.

We next assessed whether endogenous sub-14-3-3η was produced when cells were infected with coronavirus and/or enterovirus or were transfected with high molecular weight (HMW) poly (I:C), which were all reported to activate MDA5-dependent antiviral signaling (Fig 1, A, B and C). Total cell lysates from Huh7 infected with human coronavirus 229E (hCoV-229E) at 0.1 MOI were harvested from 0 to 42 hours post infection (h.p.i) and immunoblotted with anti-N and anti-14-3-3η antibodies to determine whether sub-14-3-3η was accumulated during hCoV-229E infection (Fig 1A). We observed the distinct sub-14-3-3η at 18 h.p.i accompanying with the reduced abundance of full-length 14-3-3η from 24 to 42 h.p.i. along with the accumulation of hCoV-229E N protein and cleaved PARP, one of the main cleavage targets of Caspase-3 *in vivo* (Fig 1A). Similarly, in enterovirus 71 (EV71)-infected RD cells, we could observe the accumulation of sub-14-3-3η along with the increasing expression of EV71 viral protein and cleaved PARP determined by immunoblotting (Fig 1B). In a time-course experiment of HMW poly (I:C) transfection in Huh7 cells, the inductions of MDA5 and IFIT3 protein levels over time were expected, as both genes were reported as interferon-stimulated genes (ISGs) (Fig 1C). We also observed the degradation of endogenous 14-3-3η upon the activation of Caspase-3 through time, which was indicated by the increase in the cleaved Caspase-3 abundance (Fig 1C).

To assess whether the accumulation of sub-14-3-3η was an isoform specific or a general event across the 14-3-3 family induced by MDA5 activation, we ectopically co-expressed Flag-MDA5 with several Myc-tagged 14-3-3 isoforms and examined the abundance of sub-14-3-3 proteins (Fig 1D). Distinct bands with lower molecular weight from the full-length Myc-14-3-3 isoforms, termed as sub-14-3-3γ and sub-14-3-3η, were both detected in the total cell lysates (TCL) of Flag-MDA5/Myc-14-3-3γ or Flag-MDA5/Myc-14-3-3η co-expressing cells by anti-Myc immunoblotting. After anti-Flag (MDA5) immunoprecipitation (IP), Myc-tagged full-length 14-3-3γ, 14-3-3θ, and 14-3-3η, but not 14-3-3β nor 14-3-3ε, were co-recovered with Flag-MDA5 (Fig 1D), which is consistent with our previous report [5]. Specifically, Myc-tagged sub-14-3-3η, but not Myc-tagged sub-14-3-3γ, could be strongly co-recovered with Flag-MDA5 by anti-Flag IP. Based on the abundance of full-length Myc-14-3-3η and Myc-sub-14-3-3η in the total cell lysates, the interaction between Flag-MDA5 and Myc-sub-14-3-3η

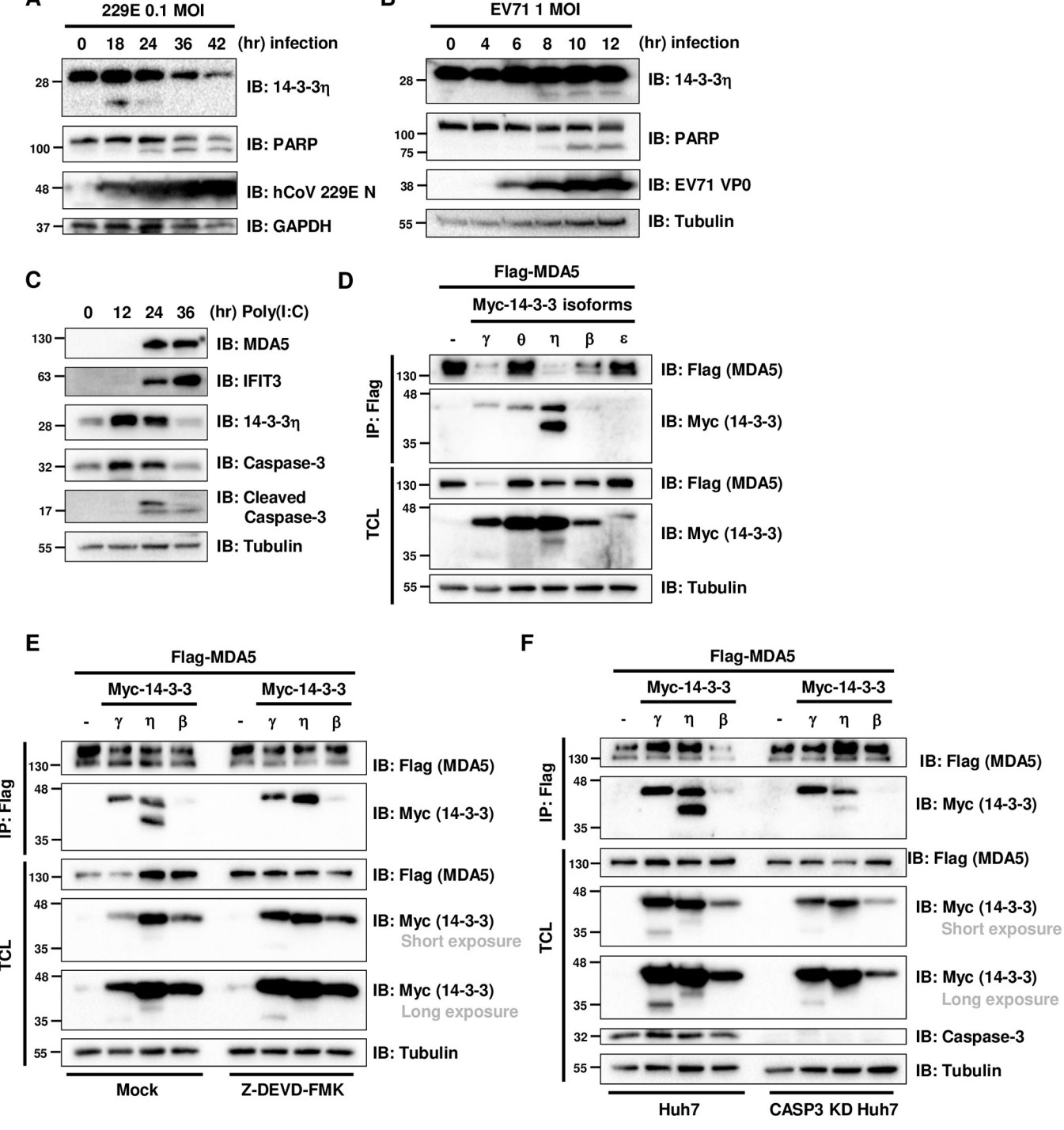

**Fig 1. The full-length 14-3-3η degradation and sub-14-3-3η accumulation observed upon MDA5 activation were associated with Caspase 3 activation.**
(A) Huh7 cells were infected with hCoV-229E at 0.1 MOI for 0 to 42 hours. Immunoblotting was performed to detect the viral protein and endogenous 14-3-3η protein abundance. (B) RD cells were infected with EV71 at 1 MOI for 0 to 12 hours. Immunoblotting was used to determine the viral infection and sub-14-3-3η accumulation. (C) Huh7 cells were transfected with 1 μg/mL HMW poly(I:C) in a time course from 0 to 36 hours. The cell lysates were analyzed by immunoblotting. (D) Huh7 cells were co-transfected with Flag-tagged MDA5 and various Myc-tagged 14-3-3 isoforms for 48 hours, followed by anti-Flag immunoprecipitation (IP). Immunoblotting was used to analyze the recovered products. (E) Huh7 cells were mock-treated or pretreated with Z-DEVD-FMK 50 μM for 1 hour and then co-transfected with Flag-tagged MDA5 and different Myc-tagged 14-3-3 isoforms for 48 hours. Anti-Flag immunoprecipitation (IP) was performed to detect the interaction of Flag-MDA5 and Myc-14-3-3 isoforms and the accumulation of Myc-sub-14-3-3. (F) Wildtype or CASP3 KD Huh7 cells were co-transfected with Flag-MDA5 and Myc-14-3-3 isoforms. After 48 hours, cells were harvested and then subjected to anti-Flag immunoprecipitation (IP) to determine the interaction of Flag-MDA5 and Myc-14-3-3 isoforms and the accumulation of Myc-sub-14-3-3 isoforms.

appeared to be stronger than that of Flag-MDA5 and Myc-14-3-3η, as more Myc-sub-14-3-3η was co-recovered with Flag-MDA5 than the full-length Myc-14-3-3η (Fig 1D). We next determined whether the accumulation and the interaction with MDA5 of sub-14-3-3η are associated with Caspase-3 activity (Fig 1E and 1F). Similarly, we ectopically co-expressed Flag-MDA5 with several Myc-tagged 14-3-3 isoforms in wildtype or Caspase-3 knock-down (CASP3 KD) Huh7 cells to detect the Myc-tagged 14-3-3 isoforms in the anti-Flag (MDA5) immunoprecipitated products. Again, we found that Myc-14-3-3γ and Myc-14-3-3η were processed to sub-14-3-3 proteins in Flag-MDA5 expressing Huh7 cells, and Myc-sub-14-3-3η was strongly co-immunoprecipitated with Flag-MDA5 by anti-Flag IP (Fig 1E and 1F). However, in Huh7 cells treated with Z-DEVD-FMK, a specific inhibitor against Caspase-3, the accumulation of Myc-sub-14-3-3η was not observed with ectopic Flag-MDA5 expression, while the accumulation of Myc-sub-14-3-3γ remained detectable (Fig 1E). This phenomenon was also observed in the CASP3 KD Huh7 cells (Fig 1F), indicating that the induction of sub-14-3-3η was associated with Caspase-3 activity.

## Sub-14-3-3η serves as negative regulator of MDA5-dependent signaling

Due to the association between Caspase-3 activity and sub-14-3-3η accumulation, we assessed whether 14-3-3η peptide sequence contains putative cleave sites of Caspase-3. The peptide sequence of human 14-3-3η (1433F_HUMAN Q04917) was analyzed by ScreenCap3 [32]. We focused on aspartate residues at the C-terminus of 14-3-3η since the Myc-14-3-3 constructs were N-terminus tagged. The returned inquiry demonstrated that D239 and D209 were the sites with the most probabilities of Caspase-3 cleavage at the C-terminus of 14-3-3η (S2A Fig). We mapped the ScreenCap3 prediction to 14-3-3η peptide sequence and found that D209 located at the very C-terminal end of αH helix, while D239 located at the C-terminal end of αI helix, of which region is also known as the beginning of 14-3-3η C-tail (Fig 2A). Based on the prediction results and the tertiary structure of 14-3-3η, we generated two Myc-tagged 14-3-3η truncation mutants by the predicted cleavage sites, Myc-14-3-3ηΔC (aa 1–239) and Myc-14-3-3ηΔαI (aa 1–209), to investigate their interactions to MDA5 and their abilities in facilitating MDA5-dependent induction in IFNβ promoter. We first investigated whether the truncation of 14-3-3η at the C-terminus HI loop generated a phenotypic mimicry of sub-14-3-3η that strongly interacts with MDA5. Full-length Myc-14-3-3η, Myc-14-3-3ηΔC, or Myc-14-3-3ηΔαI were co-expressed with Flag-MDA5 and co-immunoprecipitated by anti-Flag (MDA5) antibodies (Fig 2B). Immunoblotting result by anti-Myc showed that Myc-14-3-3ηΔαI had a similar electromobility on SDS-PAGE as sub-14-3-3η observed in the Myc-14-3-3η expressing lysates (Fig 2B). The immunoprecipitated and co-recovered products were immunoblotted by anti-Flag and anti-Myc antibodies. Compared to Myc-14-3-3η, Myc-14-3-3ηΔC did not show more binding to Flag-MDA5 than full-length Myc-14-3-3η (Fig 2B). However, the abundance of anti-Flag co-immunoprecipitated Myc-14-3-3ηΔαI was drastically increased when compared to Myc-14-3-3η (Fig 2B), indicating a stronger binding affinity to Flag-MDA5 of Myc-14-3-3ηΔαI when compared to the full-length Myc-14-3-3η. We also generated Myc-14-3-3η (1–202) and Myc-14-3-3η (1–203) terminated at D202 and/or D203, which were predicted as low potential sites for Caspase-3 cleavage (S2A Fig). We found that when αI helix of Myc-14-3-3η was deleted, the interaction toward Flag-MDA5 would be increased, as Myc-14-3-3η (1–202) and/or Myc-14-3-3η (1–203) showed the same phenotype as Myc-14-3-3ηΔαI (S2B Fig), indicating the αI helix of 14-3-3η greatly controls the interaction between 14-3-3η and MDA5.

We next determined whether 14-3-3η truncation at the C-terminus may differentially control MDA5-dependent type I IFN induction. Myc-14-3-3η, Myc-14-3-3ηΔC and Myc-14-3-3ηΔαI were ectopically co-expressed with Flag-MDA5, Flag-N-MDA5, or Myc-MAVS

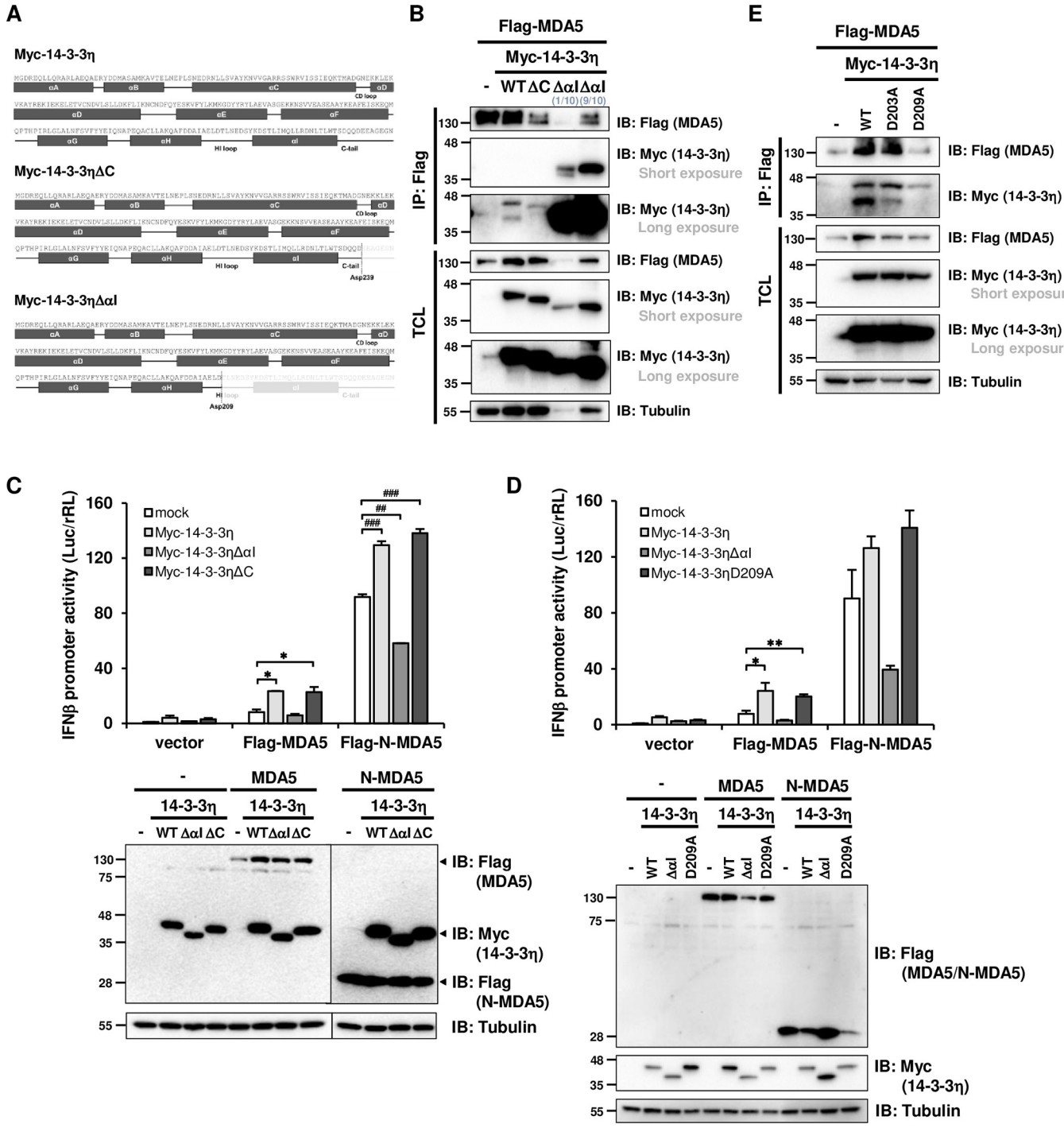

**Fig 2. D209 of 14-3-3η has a critical role in sub-14-3-3η accumulation during MDA5 activation.** (A) The constructions of Myc-tagged 14-3-3η, 14-3-3ηΔC and 14-3-3ηΔαI are showed by illustration. (B) Huh7 cells were co-transfected with Flag-MDA5 and Myc-14-3-3η wildtype or truncated mutants for 48 hours, followed by anti-Flag immunoprecipitation (IP) to determine the interactions between Flag-MDA5 and the Myc-14-3-3η constructs. (C) Indicated Myc-14-3-3η constructs were co-transfected with empty vector, Flag-MDA5 or Flag-N-MDA5 into Huh7 cells, and the cell lysates were utilized to determine the IFNβ promoter activities by dual luciferase reporter assay. *: $p<0.05$, when compared with Flag-MDA5, mock. ##: $p<0.01$, ###: $p<0.001$, when compared with Flag-N-MDA5, mock. Ectopic protein expression levels were analyzed by immunoblotting. (D) Indicated Myc-14-3-3η constructs were co-transfected with empty vector, Flag-MDA5 or Flag-N-MDA5 into Huh7 cells. The IFNβ promoter activities were measured via dual luciferase reporter assay. *: $p<0.05$, **: $p<0.01$, when compared with Flag-MDA5, mock. Ectopic protein expression levels were analyzed by immunoblotting. (E) Huh7 cells were co-transfected with Flag-MDA5 and Myc-14-3-3η wildtype or D to A mutants for 48 hours, then subjected to anti-Flag immunoprecipitation (IP) to determine the interactions of Flag-MDA5 and Myc-14-3-3η wildtype or D to A mutants.

respectively to drive IFNβ promoter activity (Figs 2C and S2C). The protein levels of ectopic expression were determined by immunoblotting. With similar expression level, the ectopic effects of Myc-14-3-3η and/or Myc-14-3-3ηΔC toward MDA5-mediated IFNβ promoter activity were similar, while the ectopic expression of Myc-14-3-3ηΔαI reduced the MDA5-mediated IFNβ promoter activity (Fig 2C); this phenotype was more pronounced in N-MDA5-mediated IFNβ promoter activity but was not observed in MAVS-mediated IFNβ promoter activity (Figs 2C and S2C), suggesting that 14-3-3ηΔαI specifically regulated MDA5-dependent antiviral signaling. Myc-14-3-3η (1–202) and/or Myc-14-3-3η (1–203) showed the same negative role as Myc-14-3-3ηΔαI in MDA5-mediated IFNβ promoter activity (S2D Fig). Loss of the 14-3-3η C-tail did not much affect the binding activity toward MDA5 (Fig 2B) nor the ability to promote MDA5-dependent IFNβ promoter activity (Fig 2C). These results indicated that 14-3-3η full-length and 14-3-3ηΔαI both bind to MDA5 but contrarily regulate MDA5 activation. The induced accumulation of sub-14-3-3η may therefore serve as a negative feedback mechanism to regulate MDA5-dependent antiviral signaling.

Since that Myc-14-3-3ηΔαI and Myc-14-3-3η (1–203) showed the same negative role in MDA5-mediated IFNβ promoter activity and that D209 and D203 were both predicted to be cleaved by Caspase 3 with high and/or low probabilities, we next generated two 14-3-3η mutants with D-to-A substitutions, including Myc-14-3-3η D209A and Myc-14-3-3η D203A. The abilities of these D-to-A mutants of 14-3-3η in facilitating MDA5-mediated IFNβ promoter activities were next assessed (S2E Fig). Same as we previously reported [5], we again observed ectopic expression of 14-3-3η enhanced MDA5-mediated IFNβ promoter activity (Fig 2D), and we found that Myc-14-3-3η D209A, was able to enhance the MDA5-mediated IFNβ promoter activity at the comparable or higher levels when compared to that of Myc-14-3-3η (Fig 2D). We then assessed whether sub-14-3-3η could still accumulated when these mutants are co-expressed with Flag-MDA5 (Fig 2E). We found that all these Myc-14-3-3η mutants were able to interact with Flag-MDA5 by anti-Flag (MDA5) co-immunoprecipitation; however, Myc-14-3-3η D209A was not able to form sub-14-3-3η, compared to that of Myc-14-3-3η (wildtype) (Fig 2E). This phenotype was not observed in Myc-14-3-3η D203A expressing cells, suggesting that D209 was the critical site for the Caspase-3 dependent sub-14-3-3η accumulation (Fig 2E).

## Sub-14-3-3η loses the ability to facilitate MDA5 redistribution to the mitochondria-MAM fraction

Based our findings, we hypothesized a negative regulatory role of sub-14-3-3η against MDA5-mediated antiviral signaling and would like to identify what is the mechanism behind. We have previously identified that the molecular mechanism of how 14-3-3η promotes MDA5-mediated type I IFN induction is through facilitating MDA5 aggregation and redistribution [5]. To confirm sub-14-3-3η may negatively regulate MDA5 activation as shown in Fig 2, we utilized 14-3-3η knock-down (KD) Huh7 cells reconstituted with Myc-tagged 14-3-3η or 14-3-3ηΔαI to investigate the expression of ISGs as a readout of type I IFN induction (Fig 3A). The immunoblotting results confirmed the 14-3-3η knock-down and Myc-tagged 14-3-3η transfection efficiencies. As known ISGs, MDA5 and IFIT3 protein levels were increased in response to HMW poly(I:C) transfection in the control Huh7 cells as detected by immunoblotting (Fig 3A). However, in the 14-3-3η KD Huh7 cells, MDA5 and IFIT3 protein abundance could not be induced by poly(I:C) transfection unless Myc-14-3-3η was reconstituted (Fig 3A). Also, when complimented with Myc-14-3-3ηΔαI, the protein abundance of MDA5 and IFIT3 could not be induced to the same level as in the Myc-14-3-3η expressing cells post-poly (I:C)

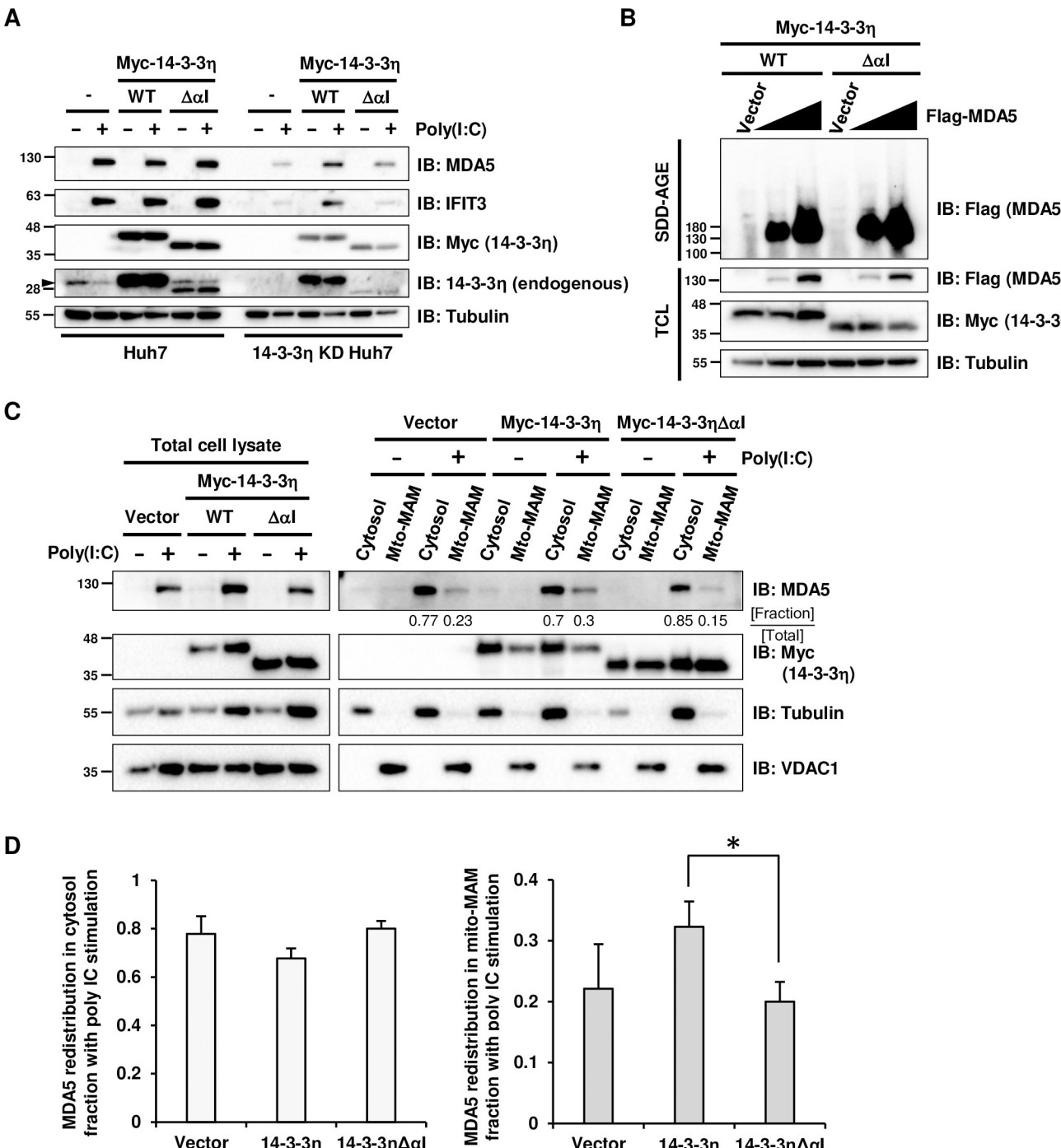

**Fig 3. Sub-14-3-3η affects MDA5 activation through inhibiting MDA5 redistribution.** (A) Wildtype and 14-3-3η KD Huh7 cells were transfected with empty vector, Myc-14-3-3η or Myc-14-3-3ηΔαI and then mock-treated or stimulated with 1 µg/mL HMW poly(I:C) for 42 hours. Cell lysates were analyzed by immunoblotting to determine the protein levels. (B) 14-3-3η KD Huh7 cells were co-transfected with increasing amount of Flag-MDA5 and Myc-14-3-3η or Myc-14-3-3ηΔαI. Cells lysed by Triton X-100 lysis buffer were analyzed by SDD-AGE to determine the MDA5 oligomerization and total protein levels were detected via SDS-PAGE. (C) Huh7 cells were transfected with empty vector, Myc-14-3-3η or Myc-14-3-3ηΔαI for 24 hours, followed by mock-transfection or 1 µg/mL HMW poly(I:C) transfection for 18 hours. Cell lysates were separated into cytosol and mito-MAM fractions. Immunoblotting was utilized for detecting the redistribution of endogenous MDA5. (D) Quantification of MDA5 band intensity in cytosol or mito-MAM fractions with HMW poly(I:C) transfection from 3 independent experiments.

transfection (Fig 3A), suggesting that Myc-14-3-3η but not Myc-14-3-3ηΔαI is able to facilitate MDA5-dependent signaling, which is consistent with our findings in Fig 2C and 2D.

Due to the strong affinity of Myc-14-3-3ηΔαI toward MDA5, we tested whether Myc-14-3-3ηΔαI may compete away full-length Myc-14-3-3η from MDA5 (S3A Fig). We co-transfected HA-MDA5, Flag-14-3-3η and/or Myc-14-3-3ηΔαI into Huh7 cells and collected the cell lysates for anti-HA (MDA5) immunoprecipitation. As expected, Flag-14-3-3η could be co-immuno-precipitated with HA-MDA5 by anti-HA antibodies (S3A Fig). When Myc-14-3-3ηΔαI was dosed into the anti-HA co-IP experiment, we overserved the interaction of Flag-14-3-3η and HA-MDA5 was reduced with low expression of Myc-14-3-3ηΔαI and was enhanced with high Myc-14-3-3ηΔαI abundance (S3A Fig), suggesting that competition between full-length and cleaved 14-3-3η to MDA5 interaction may not be the only mechanism for cleaved 14-3-3η to control MDA5 signaling. The ectopic expression of Myc-14-3-3η and/or Myc-14-3-3ηΔαI were then utilized to investigate how MDA5 aggregation and redistribution may be negatively regulated. Constant amounts of Myc-14-3-3η and/or Myc-14-3-3ηΔαI were co-transfected with increasing amounts of Flag-MDA5 into wildtype Huh7 cells (Fig 3B). The total cell lysates were analyzed for protein abundance by SDS-PAGE followed by immunoblotting, and protein oligomer or aggregation was assessed by semi-denaturating detergent agarose gel electrophoresis (SDD-AGE). We found that the aggregation or oligomerization of Flag-MDA5 was increased as the Flag-MDA5 abundance increase, and the MDA5 aggregation levels were similar in cells expressing Myc-14-3-3η and/or Myc-14-3-3ηΔαI (Fig 3B), suggesting that Myc-14-3-3ηΔαI blocked MDA5 signaling post-MDA5 aggregation.

We next assessed the MAM redistribution of MDA5 upon HMW poly (I:C) transfection, which is a critical step in MDA5-MAVS-mediated type I IFN induction. Huh7 cell lysates were fractionated into cytosol and crude mitochondria-MAM (mito-MAM) fractions, and anti-tubulin and anti-VDAC1 immunoblotting served as markers of cytosol and crude mitochondria-MAM fractions, respectively. In the vector transfected control cells, MDA5 was almost undetectable in the unstimulated cells, and under the stimulation of poly (I:C) transfection, MDA5 was primarily in the cytosol fraction and partially redistributed to the mito-MAM fraction (Fig 3C). The ratio of MDA5 in mito-MAM fraction to that in cytosol fraction was increased upon the ectopic expression of Myc-14-3-3η (Fig 3D), which is consistent with our previous reports [5]. However, in Huh7 cells expressing Myc-14-3-3ηΔαI, MDA5 redistribution to the mito-MAM fraction during poly (I:C) stimulation was less to that of the vector control cells, even that Myc-14-3-3ηΔαI itself was found in both cytosol and mito-MAM fractions (Fig 3C and 3D). Redistribution of MDA5 in response to poly(I:C) transfection was also assessed in 14-3-3η KD Huh7 cells reconstituted with Myc-tagged 14-3-3η or 14-3-3ηΔαI (S3B Fig). When transfected with poly (I:C), 14-3-3η KD Huh7 cells were unable to support MDA5 redistribution to the mito-MAM fraction, and only the reconstitution of Myc-14-3-3η but not Myc-14-3-3ηΔαI to 14-3-3η KD Huh7 cells could rescue the mito-MAM redistribution of MDA5 in poly(I:C) transfected cells (S3B Fig). Unlike full-length Myc-14-3-3η, Myc-14-3-3ηΔαI was unable to promote MDA5 redistribution (Figs 3C–3D and S3B), which may be the reason why ectopic expression of Myc-14-3-3ηΔαI impaired MDA5-dependent IFNβ promoter activity (Fig 2C and 2D).

## Cells expressing sub-14-3-3η exhibit impaired antiviral response to viral infections

To evaluate the effects of sub-14-3-3η in antiviral innate immunity, we challenged Huh7 cells expressing Myc-14-3-3η and/or Myc-14-3-3ηΔαI by Sendai Virus (SeV) Cantell strain or hCoV-229E and monitored the mRNA inductions of IFNβ. It has been reported that different

RNA viruses preferentially trigger RIG-I and/or MDA5 activation, e.g. SeV Cantell strain is known to primarily activate RIG-I while hCoV-229E activates MDA5 [33,34]. The protein expression levels of Myc-14-3-3η and/or Myc-14-3-3ηΔαI were determined by anti-Myc immunoblotting (S4A Fig). Correspondingly, the mRNA levels of IFNβ post-hCoV-229E infections were less induced in Myc-14-3-3ηΔαI-expressing Huh7 cells than those in Myc-14-3-3η-expressing Huh7 cells (Fig 4A). However, the contrary effects of Myc-14-3-3η and Myc-14-3-3ηΔαI were not observed in SeV infected cells (Fig 4A), suggesting that 14-3-3η and sub-14-3-3η specifically regulate MDA5-mediated antiviral innate immunity. The result was also confirmed by hCoV-229E infections in 14-3-3η KD Huh7 reconstituted with Myc-tagged 14-3-3η, 14-3-3ηΔαI or 14-3-3η D209A (Figs 4B and S4B). With vector transfection as a control, the induction of IFNβ mRNA after hCoV-229E infection was impaired in 14-3-3η KD Huh7 cells, and the inductions of IFNβ mRNA during hCoV-229E infection were restored in Myc-14-3-3η and Myc-14-3-3η D209A but not in Myc-14-3-3ηΔαI transfected 14-3-3η KD Huh7 (Fig 4B). Notably, reconstitution of Myc-14-3-3η D209A in 14-3-3η KD Huh7 cells showed a faster kinetics in IFNβ induction than those reconstituted with Myc-14-3-3η, while Myc-14-3-3η (wildtype) reconstituted cells showed a higher induction of IFNβ with a slower kinetics than those in Myc-14-3-3η D209A reconstituted 14-3-3η KD Huh7 cells (Fig 4B). We also assessed the hCoV-229E virus RNA levels in wildtype or 14-3-3η KD Huh7 cells, and we found that Huh7 cells expressing Myc-14-3-3ηΔαI were slightly more permissive to the hCoV-229E infection, as the vRNA levels at both 6 and 12 hours post-hCoV-229E infections were the highest among all (Figs 4C and S4C). The reconstitution of Myc-14-3-3ηΔαI in 14-3-3η KD Huh7 cells showed comparable vRNA levels during the hCoV-229E infection, whereas the reconstitution of Myc-14-3-3η D209A provided a better inhibition to virus replication at both 6 and 12 hours post-hCoV-229E infections (Fig 4C). All these phenotypes by reconstituting different 14-3-3η mutants could not be observed in MDA5 KD Huh7 cells (Fig 4D), suggesting that the effects of sub-14-3-3η or 14-3-3η D209A were dependent on the expression and activation of MDA5.

As we have observed the accumulation of sub-14-3-3η during hCoV-229E infection (Fig 1A), we assessed whether the phenotype was associated with Caspase-3 activation in mock- and/or Caspase-3 inhibitor-treated Huh7 cells (Fig 4E). Indeed, sub-14-3-3η abundance peaked at 18 hours post-infection and faded away as shown by anti-14-3-3η immunoblotting (Fig 4E). We also observed reduced full-length 14-3-3η protein abundance when the infection time prolonged. The accumulation of cleaved PARP and activated Caspase-3 were also observed. However, protein lysates from hCoV-229E infection in Huh7 cells treated with Z-DEVD-FMK, a Caspase-3 specific inhibitor, showed decrease abundance of sub-14-3-3η and cleaved PARP, along with absence of cleaved Caspase-3 by immunoblotting (Fig 4E). These results and our findings in Fig 2 suggested that Caspase-3 is activated during hCoV-229E infection, and active Caspase-3 is largely responsible for the formation of sub-14-3-3η.

We further tested whether this phenotype could also be observed in EV71 infected Huh7 cells and compared the protein abundance of sub-14-3-3η, cleaved PARP, and cleaved Caspase-3 between Huh7 and CASP3 KD (Caspase-3 knock-down) Huh7 cells (Fig 4F). Again, we observed cleaved Caspase-3, cleaved PARP and sub-14-3-3η during EV71 infection; however, in the CASP3 KD Huh7 cells, although cleaved Caspase-3 was not detected by immunoblotting as expected, we did not see obvious alterations in protein abundance of sub-14-3-3η and cleaved PARP between CASP3 KD Huh7 cells and control Huh7 cells (Fig 4F), suggesting that certain RNA virus infection may activate multiple pathways to induce the formation of sub-14-3-3η to ensure the inactivation of 14-3-3η and MDA5. Nevertheless, several RNA viruses are known to induce apoptotic Caspases activities independent from MDA5 activation. Here we showed that the ectopic expression of EV71 viral protease 2A, but not its protease-inactive

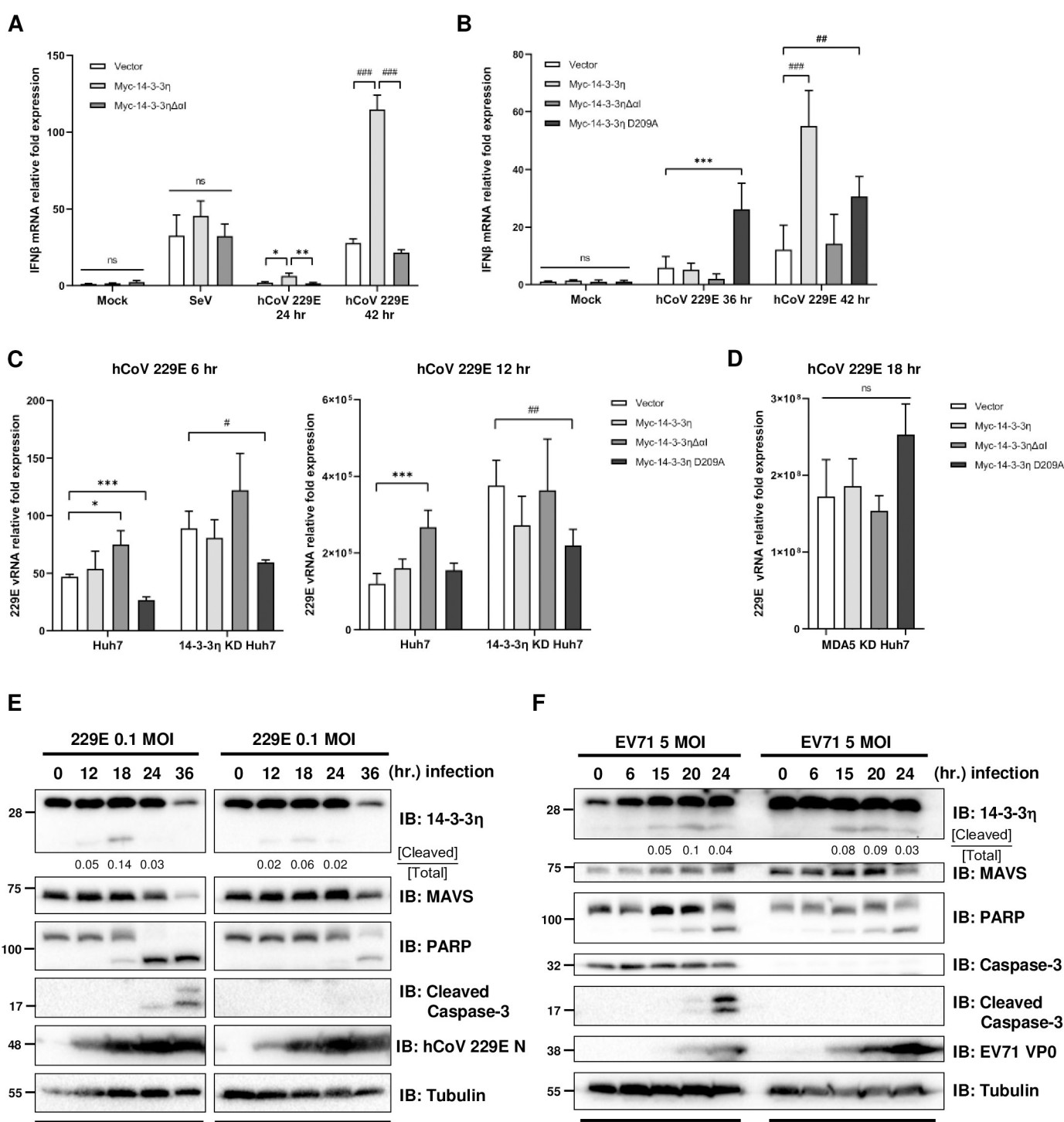

**Fig 4. Cells expressing sub-14-3-3η are more permissive to viral infections.** (A) Huh7 cells were transfected with empty vector, Myc-14-3-3η or Myc-14-3-3ηΔαI for 24 hours and subsequently mock-treated or infected with SeV at 100 HAU for 24 hours or hCoV-229E at 1 MOI for 24 or 42 hours. Quantitative RT-PCR was performed to analyze the mRNA expression levels of IFNβ. (*: p<0.05, **: p<0.01; ###: p<0.001) (B) Empty vector, Myc-14-3-3η, Myc-14-3-3ηΔαI or Myc-14-3-3η D209A were transfected into 14-3-3η KD Huh7 cells, followed by mock-treated or infected with hCoV-229E at 1 MOI for 36 or 42 hours. The relative mRNA expression levels of IFNβ were analyzed by quantitative RT-PCR. (***: p<0.001; ##: p<0.01, ###: p<0.001) (C) Empty vector, Myc-14-3-3η, Myc-14-3-3ηΔαI or Myc-14-3-3η D209A were transfected into wildtype or 14-3-3η KD Huh7 cells followed by mock-treated or infected with hCoV-229E at 0.01 MOI for 6 or 12 hours, and the intracellular vRNA levels were determined by quantitative RT-PCR. (*: p<0.05, ***: p<0.001; #: p<0.05, ##: p<0.01) (D) Empty vector, Myc-14-3-3η, Myc-14-3-3ηΔαI or Myc-14-3-3η D209A were transfected into MDA5 KD Huh7 cells followed by mock-treated or infected with hCoV-229E at 0.01 MOI for 18 hours, and the intracellular vRNA levels were determined by quantitative RT-PCR. (E) Huh7 cells were mock-treated or pretreated with

Z-DEVD-FMK 50 μM for 1 hour and subsequently infected with hCoV-229E at 0.1 MOI for 0 to 36 hours. Immunoblotting was performed to detect the viral infection and other protein levels. (F) Wildtype and CASP3 KD Huh7 cells were infected with EV71 at 5 MOI for 0 to 24 hours, and immunoblotting was performed to determine the viral infection and other protein levels.

mutant H21L, indeed induced apoptotic Caspases activation and reduced full-length 14-3-3η expression (S4D Fig), suggesting that RNA viruses may have multiple strategies in controlling 14-3-3η-mediated MDA5-dependent antiviral signaling.

## Discussion

For a long time, MDA5 has been known as an important PRR which raises the alarm and triggers the type I IFN induction pathway against the invading RNA viruses [35,36]. Comparing to the auto-suppressive mechanism of RIG-I, which is another member of the RLR family, how MDA5 activities are controlled still requires further investigation. Recently, it was demonstrated that 14-3-3η formed complex with MDA5 and played a key regulatory role in MDA5 activation which enhanced the MDA5 oligomerization and redistribution [5]. While investigating the role of 14-3-3η in MDA5 activation, we noticed that in addition to the full-length 14-3-3η, a 14-3-3η sub-isoform was able to interact with MDA5 even with high affinity (Figs 2B and S2B). Accordingly, we proposed that to prevent the overactivation of MDA5, a negative feedback was achieved by 14-3-3η sub-isoform to temporally regulate the MDA5-mediated type I IFN induction pathway. These results suggested that once type I IFN signaling was activated to a certain degree, the apoptosis related Caspases would be triggered to clean away the infected and/or damaged cells as well as to prevent the constitutive activation of MDA5 and over-inflammation. The apoptotic Caspases, especially Caspase-3, would mediate the cleavage of 14-3-3η to produce the sub-14-3-3η. Although the truncated form of 14-3-3η lacking the αI helix did not show dominant-negative effect toward MDA5 activation (Figs 2 and 4), due to the strong interaction between sub-14-3-3η and MDA5, we proposed that sub-14-3-3η would occupy the MDA5 and prevent MDA5 interaction to the full-length 14-3-3η and therefore reduce the mitochondrial redistribution of MDA5 to interact with MAVS. This hypothesis was partly supported by our results (S3A Fig) that the addition of sub-14-3-3η may interrupt the interaction between MDA5 and full-length 14-3-3η. The other possibility is that sub-14-3-3η may strongly bind to N-MDA5 and thus prevent MAVS from interacting with the CARDs of MDA5. Via producing sub-14-3-3η to inhibit the MDA5 activities, cells can return to homeostasis when the infection is resolved. This regulation is particularly important in MDA5-dependent signaling, as MDA5 can be activated when the intracellular abundance of MDA5 is high [5]. Further studies have indicated that cellular RNAs play a significant role as PAMPs in activating the RLR–MAVS signaling pathway. Although the specific host ligands for MDA5 remain less understood, MDA5 can become active when mitochondrial RNA degradation is impaired, allowing mitochondrial double-stranded RNA to escape into the cytoplasm [37]. Additionally, the expression of endogenous retroviral elements can trigger the MDA5–MAVS signaling axis to express type I IFNs [38,39]. Our findings of Caspase-3 dependent sub-14-3-3η accumulations observed during acute RNA viral infections may also contribute to the control of MDA5 deactivation under sterile inflammation.

The emerging roles of chaperone proteins in antiviral innate immunity have been revealed in the past decade. In previous reports, members of 14-3-3 protein family were shown to be important for cells to maintain normal functions by manipulating the activities of various signaling pathways, and in the RLR-related type I IFN induction pathway, MDA5 was not the only RLR that is regulated by 14-3-3 chaperone protein [7,40]. 14-3-3ε was showed to promote the RIG-I/TRIM25/14-3-3ε translocon formation and therefore transduced the active

signaling to MAVS [7]. RIG-I and MDA5 are both critical to respond to RNA virus infections, and their activities require to be well controlled to prevent overactive inflammation. In this study, we proposed the negative feedback in the regulated relation between MDA5 and 14-3-3η. As RIG-I and MDA5 both contain the Caspase activation and recruitment domains (CARDs) at their N-terminus, it is intriguing to determine if there is a similar mechanism between RIG-I and 14-3-3ε, especially when 14-3-3ε is already showed to be substrate of Caspase-3 [28]. In previous reports, 14-3-3ε had decreased binding affinity to its target protein, Bad, after Caspase-3 cleaved 14-3-3ε at Asp238 and removed partial C-tail [28]. While preparing this report, a study revealed that EV71 3C protease cleaved 14-3-3ε at a site close to a known Caspase-3 cleavage site at the C-tail, and the cleavage impaired the ability of 14-3-3ε to interact with RIG-I [41]. It is fascinating to know whether it is a common regulatory method for cells to mediate cleavage on other 14-3-3 proteins by Caspase-3 after infection for limiting the overactivation of RLR-signaling.

Nevertheless, previous reports have shown that to prevent the induction of type I IFN, DENV NS3 and ZIKV NS3 can compete 14-3-3ε with RIG-I and/or 14-3-3η with MDA5 [30,42]. It is intriguing whether certain viral infections may promote the formation sub-14-3-3η to antagonize MDA5 signaling. For instance, it has been reported that enterovirus infection induced apoptotic Caspases, including Caspase-3, to cleave MDA5 at the C-terminus and subsequently dampened the induction of type I IFN [43]. It was not fully understood how the cleaved MDA5 during enterovirus infection, which remained the intact N-terminus CARDs, would lose its activity to promote IFNβ induction. Our report provides another aspect to explain the phenotype observed in previous study, which is that Caspase-3 activation during EV71 infection targets not only MDA5 but also 14-3-3η to antagonize type I IFN induction. Our results suggested that Caspases other than Caspase-3 may also be involved in the formation of sub-14-3-3η, as in CASP3 KD Huh7 cells, sub-14-3-3η was still observed during EV71 infection (Fig 4F). In fact, expressions of many viral proteins without the full-course of infection were also able to trigger apoptotic Caspases activation [22]. Caspases involved in apoptosis, notably Caspase-3 and Caspase-7, regulate the production of excessive interferons by cleaving several essential proteins, such as cGAS in cytosolic DNA sensing and MAVS in cytosolic RNA sensing [20]. Moreover, IRF3, a shared downstream element in both pathways, can be targeted for cleavage by both Caspase-3 and Caspase-8 [44,45]. Additionally, Caspase-8 modulates the activation of IFN pathways by directly interacting with components of the RIG-I signaling complex, thereby preventing their excessive activation [46]. It may require further *in vitro* cleavage analysis of 14-3-3η with a panel of purified Caspases in the presence or absence of MDA5 and its RNA ligands to elucidate the role of Caspase 3 directly. We also believe that there are other viral proteins that may potentially induce sub-14-3-3η accumulation to control antiviral innate immunity. In fact, the peptide sequences of 14-3-3η across different species are highly conserved (S4E Fig), and therefore targeting 14-3-3η as an antagonistic mechanism to downregulate antiviral innate immunity might also be evolutionarily conserved. In summary, we proposed a model to describe sub-14-3-3η cleaved by Caspase-3 as a negative feedback of MDA5-dependent signaling (Fig 5). These findings might provide inspirations to discover more in the field of antiviral innate immunity.

## Materials and methods

### Cells

Human hepatoma cell lines, Huh7 (JCRB No. 0403), and rhabdomyosarcoma (RD) cells (ATCC: CCL-136) were cultured in Dulbecco's modified Eagle medium (DMEM) supplemented with 10% fetal bovine serum (FBS). Huh7 was validated by DSMZ STR Profile

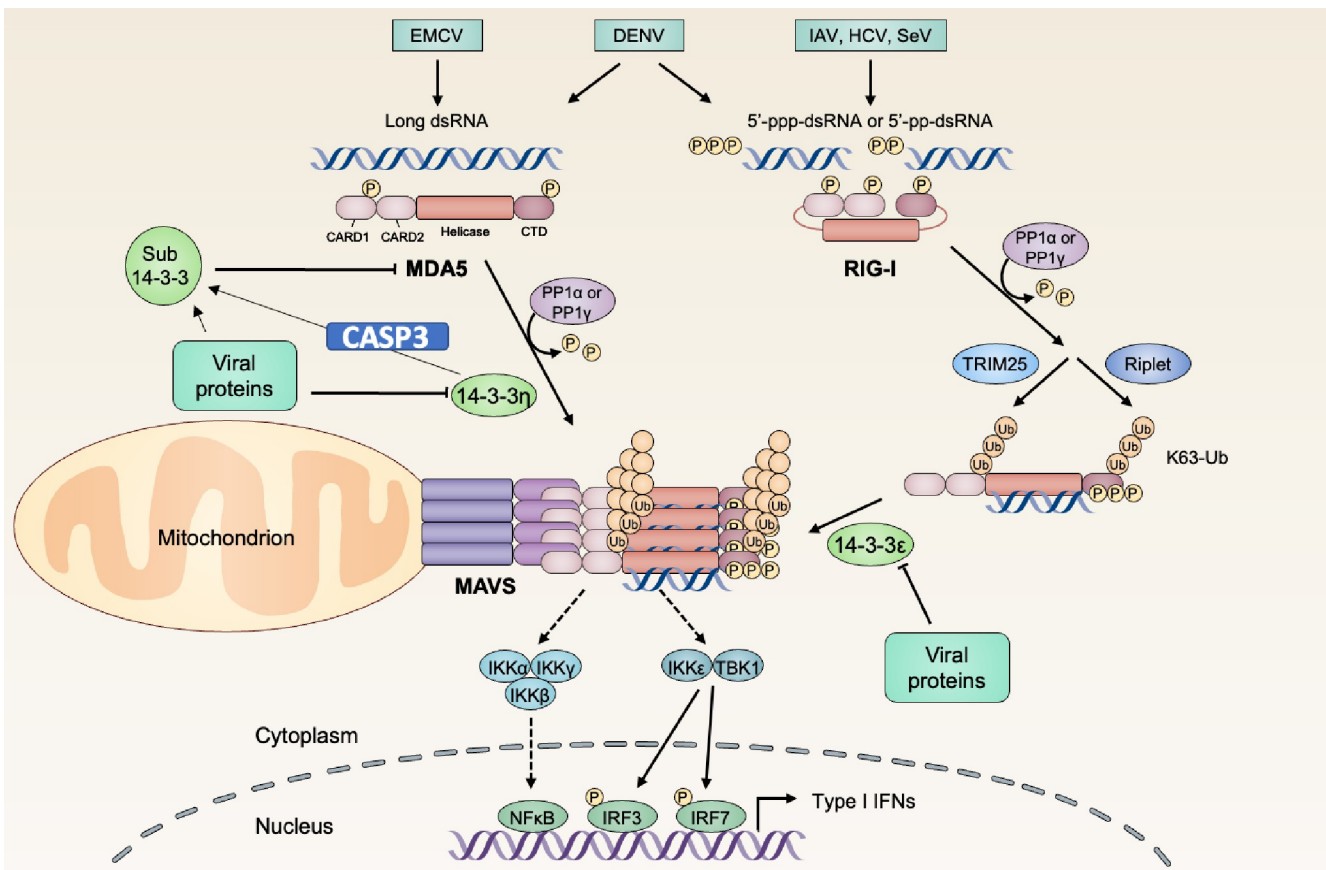

**Fig 5. Illustration of the proposed model during viral infection.** Viral infection triggers the activation of MDA5 and RIG-I anti-viral signaling. When MDA5-mediated type I IFN induction is activated, it will lead to the activation of Caspase-3. Then, 14-3-3η is cleaved by active Caspase-3, which results in the formation of sub-14-3-3η. Sub-14-3-3η competes with full-length 14-3-3η to interact with MDA5. Subsequently, MDA5 bound with sub-14-3-3 losses IFNβ-inducing function. Therefore, by forming sub-14-3-3η to block MDA5 signaling as a negative feedback, over-activation of type I IFN induction can be prevented in cells.

Database (DSMZ Number: JCRB0403) with EV 1.00. Huh7 cells were transfected with shRNA plasmids, including CASP3 shRNA plasmid from RNAiCore at Academia Sinica and 14-3-3η shRNA plasmid from Oligoengine, and then selected by puromycin at 2 μg/mL to respectively generate Caspase-3 knock-down (KD) Huh7 cells and 14-3-3η KD Huh7 cells. DMEM medium supplemented with 10% FBS and 1 μg/mL puromycin was used to maintain the KD Huh7 cells.

## Viruses infection, high molecular weight poly(I:C) stimulation and inhibitor treatment

Huh7 cells were infected with human coronavirus-229E (hCoV-229E) at a 0.01, 0.1 or 1 MOI in serum-free medium and incubated at 37°C for one hour. After that, the cells were washed by PBS and then incubated in DMED containing 2% FBS. To infect Huh7 cells with Sendai virus (SeV), cells were exposed to 100 HA unit (HAU) of SeV in DMEM supplemented with 10% FBS at 37°C. For enterovirus 71 (EV71) infection, RD cells or Huh7 cells were respectively infected with EV71 at 1 or 5 MOI in serum-free medium and incubated at 37°C for one hour. Cells were then washed with PBS and incubated in DMEM with 2% FBS. In addition, high molecular weight poly(I:C) (Invitrogen) were transfected into Huh7 cells by using TransIT-

mRNA Transfection Kit (Mirus). The Caspase-3 inhibitor Z-DEVD-FMK (MCE, 50 μM) was used for pretreating cells to inhibit the activation of Caspase-3.

## Plasmids and constructs

The constructs of Flag-tagged MDA5 and Myc-tagged 14-3-3 isoforms have been previously described [5,7]. Myc-14-3-3η ΔαI, ΔC, 1–202, 1–203, D202A, D203A, and D209A constructs were generated by site-directed mutagenesis using the QuikChange Lightning Site-Directed Mutagenesis Kit (Aligent), following the manufacturer's instructions for the detailed protocols.

## Transfection and dual luciferase IFNβ reporter assay

Fugene 6 (Promega) was used to transfect plasmids into Huh7 cells. The detailed protocols were based on the instructions of manufacturer. For dual luciferase IFNβ reporter assay, pIFNβ-Luc, pCMV-Renilla-Luc, and other plasmids were transfected into Huh7 cells for 48 hours. After harvesting and lysing the cells, the luciferase activities of IFNβ promoter and Renilla were monitored via Dual-Luciferase Assay Kit (Promega), according to the protocols in manufacturer's instructions.

## Immunoblot analysis and immunoprecipitation

Cell pellets were washed with PBS and then lysed by cold RIPA buffer (150 mM NaCl, 50 mM Tris-HCl pH = 7.5, 5 mM EDTA, 1% NP-40, 0.5% sodium deoxycholate, 0.1% SDS) containing a protease inhibitor cocktail (Roche) for 20 minutes. The cell lysates were centrifuged to remove the cell debris to obtain the protein samples. The protein samples were then mixed with 4X SDS sample buffer (240 mM Tris pH = 6.8, 5% β-mercaptoethanol, 8% SDS, 40% glycerol, bromophenol blue) and subjected to SDS-PAGE analysis. For immunoprecipitation, the protein samples were incubated with ANTI-FLAG M2 Affinity Gel at 4°C for 7 hours. The immunocomplexes were wash 3 times with cold RIPA buffer and then resuspended in 2X SDS sample buffer for SDS-PAGE analysis. The following antibodies were used in this studies: FLAG M2-HRP (Sigma-Aldrich, A8592), Myc tag (BETHYL, A190-205A), MDA5 (Cell Signaling Technology, #5321), 14-3-3η (Cell Signaling Technology, #5521; Invitrogen, #PA5-98163), VDAC1 (Abcam, ab15895), Caspase-3 (Cell Signaling Technology, #9662), cleaved Caspase-3 (Cell Signaling Technology, #9664), PARP (Cell Signaling Technology, #9542), Tubulin (Sigma-Aldrich, ZMS1039), IFIT3 (Abcam, ab76818), GAPDH (Cell Signaling Technology, #2118), hCoV-229E nucleocapsid (SinoBiological, 40640-T62), EV71 (Sigma-Aldrich, MAB979).

## Semi-denaturing detergent agarose gel electrophoresis (SDD-AGE)

The experiment procedures were previously described [5]. Cell pellets were washed with PBS and then lysed in a cold Triton X-100 lysis buffer (50mM Tris pH = 7.5, 137 mM NaCl, 1.5 mM MgCl$_2$, 1 mM EDTA, 1% Triton X-100) supplemented with a protease inhibitor (Roche) for 20 minutes. The cell lysates were centrifuged to remove the cell debris to obtain the protein samples. Each protein sample was incubated in 4X SDD-AGE sample buffer (2X TBE, 4% glycerol, 8% SDS, bromophenol blue) and then separated by using 1.5% agarose gel contained 0.1% SDS in 1X SDD-AGE running buffer (1X TBE and 0.1%SDS) at 4°C, 80V for 80 minutes. The proteins were then transferred to the NC membrane by capillary action. The NC membrane was blocked in TBST buffer with 5% bovine serum albumin (BSA) overnight.

## Cell fractionation assay

Mitochondrial/Cytosol Fractionation Kit (BioVision) was used to isolate the cellular mito-chondrial-MAM fraction from cytosolic fraction. Cell pellets were suspended in 1X cytosolic extraction buffer and incubated on ice for 10 minutes. Next, the cell lysates were homogenized by G25 needles and then centrifuged at 700 x g for 10 minutes at 4˚C. The supernatants were collected and centrifuged again. Afterward, the supernatants underwent centrifugation at 1000 x g for 30 minutes at 4˚C to isolate the mitochondrial-MAM fraction from cytosolic fraction. The supernatants were collected as cytosolic fraction and the pellets were resuspended with 1X mitochondrial extraction buffer as the mitochondrial-MAM fraction.

## Quantitative real-time PCR

RNA was isolated from cells by using TRIzol (Invitrogen). The iScript cDNA synthesis kit (Bio-Rad) was used for RNA reverse transcription. The cDNA products were subjected to qPCR with SYBR Green by using StepOne Real-Time PCR system (Applied Biosystems). The primer sequences are as following: Ifnb (Fwd: CTTTCCATGAGCTACAACTTGC, Rev: CATTCAATTGCCACAGGAGC), hCoV-229E (Fwd: TGGCCCCATTAAAAATGTGT, Rev: CCTGAACACCTGAAGCCAAT), and hGAPDH (Fwd: CCACATCGCTCAGACACCAT, Rev: AAAAGCAGCCCTGGTGACC).

## Statistical analysis

All the results were shown as mean standard deviation (SD), and the experiments were repeated at least three times. The two-tailed Student's t-test was used for analyzing the qPCR and luciferase reporter assay, $*p < 0.05$, $**p < 0.01$, and $***p < 0.001$.

## Supporting information

**S1 Fig. The accumulation of sub-14-3-3η was observed upon MDA5 expression.** (A) The schematic diagram of different MDA5 constructs. (B) Myc-MDA5 1–889 was co-transfected with empty vector or Flag-14-3-3η into Huh7 cells, followed by the anti-Myc immunoprecipi-tation (IP) to determine the Flag-sub-14-3-3η accumulation.
(TIF)

**S2 Fig. Prediction of the Caspase-3 cleavage sites of 14-3-3η upon MDA5 activation.** (A) Prediction of Caspase-3 cleavage sites of 14-3-3η via ScreenCap3 database. The probability score ranged from 0 to 1. The higher score meant the more probability of the Caspase-3 cleav-age. (B) Huh7 cells were co-transfected with Flag-MDA5 and Myc-14-3-3η wildtype or trun-cated mutants for 48 hours, followed by anti-Flag immunoprecipitation (IP) to determine the interactions of Flag-MDA5 and Myc-14-3-3η wildtype or truncated mutants. (C) Indicated Myc-14-3-3η constructs were co-transfected with empty vector or Myc-MAVS into Huh7 cells and then cell lysates were detected the IFNβ promoter activities by dual luciferase reporter assay. (D) to (E) Myc-14-3-3η wildtype, (D) truncated mutants, or (E) indicated D to A mutants were co-transfected with empty vector or Flag-MDA5 into Huh7 cells. Dual luciferase reporter assay was performed to monitor the IFNβ promoter activities of cell lysates.
(TIF)

**S3 Fig. The molecular mechanism of how sub-14-3-3η may affect MDA5 activation.** (A) HA-MDA5, Flag-14-3-3η and Myc-14-3-3ηΔαI were co-transfected into Huh7 cells. Anti-HA immunoprecipitation (IP) was performed to determine the interaction of the ectopic express-ing proteins. (B) 14-3-3η KD Huh7 cells were transfected with empty vector, Myc-14-3-3η or

Myc-14-3-3ηΔαI for 24 hours, followed by mock-transfection or 1 μg/mL HMW poly(I:C) transfection for 18 hours. Cell lysates were separated into cytosol and mito-MAM fractions. Immunoblotting was utilized for detecting the redistribution of endogenous MDA5.
(TIF)

**S4 Fig. Viral protease induced apoptosis promoted the degradation of 14-3-3η.** (A) Continued to Fig 4A, immunoblotting was used to confirm the ectopic expression. (B) Continued to Fig 4B, immunoblotting was used to confirm the ectopic expression. (C) Continued to Fig 4C and 4D, immunoblotting was utilized to confirm the ectopic expression in Huh7 cells, 14-3-3η KD Huh7 cells and MDA5 KD Huh7 cells. (D) Constant amount of Myc-14-3-3η and empty vector, EV71 2A-eGFP or EV71 2A H21L-eGFP were co-transfected into Huh7 cells for 72 hours. The protein levels were analyzed by immunoblotting. (E) Whole amino acid residues of 14-3-3η from different species were aligned. The sequences of 14-3-3η across different species were highly conserved.
(TIF)

**S1 Raw dataxlsx. Raw data of experimental values behind the means and standard deviations used to build graphs.**
(XLSX)

## Acknowledgments

We thank for Dr. Rei-Lin Kuo for providing EV71 2A constructs and insightful discussions.

## Author Contributions

**Conceptualization:** Helene Minyi Liu.

**Data curation:** Yun-Jui Chan, Nien-Tzu Liu, Helene Minyi Liu.

**Formal analysis:** Yun-Jui Chan.

**Funding acquisition:** Helene Minyi Liu.

**Investigation:** Yun-Jui Chan, Nien-Tzu Liu, Fu Hsin, Jia-Ying Lu, Jing-Yi Lin, Helene Minyi Liu.

**Methodology:** Yun-Jui Chan, Nien-Tzu Liu, Jia-Ying Lu, Jing-Yi Lin, Helene Minyi Liu.

**Project administration:** Helene Minyi Liu.

**Resources:** Jing-Yi Lin.

**Supervision:** Helene Minyi Liu.

**Validation:** Yun-Jui Chan, Nien-Tzu Liu.

**Visualization:** Fu Hsin.

**Writing – original draft:** Yun-Jui Chan, Nien-Tzu Liu, Helene Minyi Liu.

**Writing – review & editing:** Yun-Jui Chan, Helene Minyi Liu.

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
