## [Decision Letter · Decision Letter 0]

26 Feb 2024

Dear Dr. Liu,

Thank you very much for submitting your manuscript "Temporal regulation of MDA5 inactivation by Caspase-3 dependent cleavage of 14-3-3η" for consideration at PLOS Pathogens. As with all papers reviewed by the journal, your manuscript was reviewed by members of the editorial board and by several independent reviewers. In light of the reviews (below this email), we would like to invite the resubmission of a significantly-revised version that takes into account the reviewers' comments.

We cannot make any decision about publication until we have seen the revised manuscript and your response to the reviewers' comments. Your revised manuscript is also likely to be sent to reviewers for further evaluation.

Sincerely,

Chia-Yi Yu, Ph.D.

Guest Editor

PLOS Pathogens

Michael Letko

Section Editor

PLOS Pathogens

Michael Malim

Editor-in-Chief

PLOS Pathogens

orcid.org/0000-0002-7699-2064

Thank you for submitting your manuscript to PLoS Pathogens. After a thorough review by two referees, we have reached the decision to request a major revision of your manuscript.

Cleavage of 14-3-3η by Caspase-3 during MDA5 activation produces sub-14-3-3η, impairing antiviral innate immunity, suggesting an interesting hypothesis by which RNA viruses may antagonize host defenses by activating cell death. Both innate immune responses and death are important issues for a host cell response to invading pathogens and contribute to the infection's final outcomes. From a pathogen's point of view, innate immunity should be turned off first so that makes the infection and the cause of death easier. Does the virus really benefit from sub-14-3-3 to its infection? Why and how a pathogen wants to use a relatively late event of the infection (caspase activity) to deal with a relatively urgent issue (antagonizing innate immunity) is interesting and of which biological significance needs more discussion. Meanwhile, showing 14-3-3 as a direct substrate of the caspase is suggested to further strengthen the statement.

Upon reviewing the attached reports, you will note that the reviewers have raised concerns about the study and manuscript preparation that hampered the acceptance of this manuscript at the current stage. Despite we hope to see experiments showing that 14-3-3 cleavage does have an impact on virus infection, a biological explanation and discussion should be included, no matter whether the impact existed or not. Besides, scientific proofreading and grammar copyediting were also suggested. Consequently, we kindly request that you address these concerns through additional work.

Reviewer's Responses to Questions

**Part I - Summary**

Reviewer #1: Chan et al. describe the identification of a cleaved form of 14-3-3η and its plausible role in regulating MDA5 function. In a previous report, the authors have noted that a distinct band of 14-3-3η with faster electromobility, termed sub-14-3-3η, was observed in a co-immunoprecipitation assay of MDA5. In the current study, Chan et al. found that the expression of sub-14-3-3η was specific to the presence of caspase-3 and was upregulated by coronavirus and enterovirus infections. In addition, overexpression of sub-14-3-3η was shown to downregulate MDA5-mediated IFN induction and subsequent ISG expression, and the authors claimed that this was likely due to inhibition of MDA5 redistribution to mitochondria. Consistent with these findings, reduced IFN induction in response to virus infection was also observed in sub-14-3-3η-expressing cells.

The experiments presented were performed appropriately and the manuscript is easy to follow. The identification of a possible negative regulator of MDA5, which is generated from full-length 14-3-3η by Caspase-3, is novel. From an immunological point of view, the study would be interesting in terms of auto-feedback regulation of the innate immune response. However, with the data presented, the functional role of sub-14-3-3η in virus replication remains unclear.

Reviewer #2: In this study, Chan et al. observed that 14-3-3n antibody stains a smaller band of the protein upon virus infection. In their prior work from 2019, they describe the role of 14-3-3n as a chaperone for MDA5. In this work, they find that after MDA5 activation, 14-3-3n is cleaved into a smaller molecular weight isoform that inhibits MDA5 function. They find that the cleavage depends on caspase-3 and residue D209. They then describe their finding that sub-14-3-3n exerts its activity in part by preventing MDA5 from localizing to the MAM, and this allows viruses that activate MDA5 to replicate better. From a host standpoint, sub-14-3-3u may be part of a negative feedback loop to prevent runaway inflammation.

Overall, the work and data presentation as figures and legends are clear. The manuscript has many typos and grammatical errors that sometimes interfere with the presentation of ideas.

**Part II – Major Issues: Key Experiments Required for Acceptance**

Reviewer #1: 1. The association of MDA5 with sub-14-3-3η and the redistribution of MDA5 should also be demonstrated by immunofluorescence analysis.

2. The weakest point of this paper is that it does not show a functional role for sub-14-3-3η in viral replication, thus one should question the submission to PLoS Pathogens. If sub-14-3-3η functions as a negative regulator of MDA5-mediated IFN induction even in virus-infected cells, expression of the caspase-3-resistant version of 14-3-3η (i.e. the D209A mutant) confers a higher antiviral property than the WT one. Also, virus replication should be enhanced by sub-14-3-3η expression. These data are crucial, I believe.

3. Page 7, 1st line "a stronger binding affinity...": The data in Fig. 2B do not clearly demonstrate that the affinity of sub-14-3-3η for MDA5 is "stronger" than that of full-length 14-3-3η. A competition assay should be performed in which full-length and sub-14-3-3η are simultaneously transfected, followed by MDA5 immunoprecipitation.

Reviewer #2: Figure 2E it is very hard to tell that there is no sub-14-3-3η in the D209A lane due to the dark spots there. Possible to repeat, or maybe try probe/re-probing with 14-3-3 ab?

**Part III – Minor Issues: Editorial and Data Presentation Modifications**

Reviewer #1: 1. Molecular weight mass should be indicated on all immunoblotts.

2. Fig. 2E: Immunoblot of immunoprecipitated D209A by Myc antibody detection is unclear.

3. Where is Fig.3D mentioned in the main body of text?

3. Fig. 4A and B lack statistical analysis.

4. Page 10, line 3 "MycMycCorrespondingly," and line 4 "MycMyc-14-3-3η": Please correct.

5. Page 6, "Sub-14-3-3η serves as ...." section, first paragraph: It would be informative to indicate the putative cleavage site to generate sub-14-3-3η is indicated using a tertiary structure of 14-3-3η.

Reviewer #2: Text throughout requires some copy editing to improve tenses and typos.

Introduction: grammar. Need to introduce 14-3-3 family and members better (just ε

and η?

Figure 1C based on introduction, if cleavage of 14-3-3η is part of normal negative feedback, shouldn’t there be some cleaved 14-3-3η on a long exposure?

Figure 1D. On IB for Flag, why is MDA5 level different in TCL and in IP: Flag across conditions?

Figure 2C where is myc-MAVS? Mentioned in text but not figure or legend. Only in S2C?

It would be helpful, wherever possible, to minimize chopping up the blots.

Discussion: recommend a bit more robust discussion on other possible mechanisms of sub-14-3-3n inhibition of MDA5

PLOS authors have the option to publish the peer review history of their article (what does this mean?). If published, this will include your full peer review and any attached files.

Reviewer #1: No

Reviewer #2: No
---

## [Decision Letter · Decision Letter 1]

23 May 2024

Dear Dr. Liu,

We are pleased to inform you that your manuscript 'Temporal regulation of MDA5 inactivation by Caspase-3 dependent cleavage of 14-3-3η' has been provisionally accepted for publication in PLOS Pathogens.

Best regards,

Chia-Yi Yu, Ph.D.

Guest Editor

PLOS Pathogens

Michael Letko

Section Editor

PLOS Pathogens

Michael Malim

Editor-in-Chief

PLOS Pathogens

orcid.org/0000-0002-7699-2064

Reviewer Comments (if any, and for reference):

Reviewer's Responses to Questions

**Part I - Summary**

Reviewer #1: (No Response)

Reviewer #2: I appreciate the authors' efforts in addressing the reviewers' concerns and think these have improved the paper, particularly better describing the findings and context. The authors addressed my concerns. Generally speaking, still needs some copy editing. I do think the data provided support the study conclusions.

**Part II – Major Issues: Key Experiments Required for Acceptance**

Reviewer #1: (No Response)

Reviewer #2: (No Response)

**Part III – Minor Issues: Editorial and Data Presentation Modifications**

Reviewer #1: (No Response)

Reviewer #2: (No Response)

PLOS authors have the option to publish the peer review history of their article (what does this mean?). If published, this will include your full peer review and any attached files.

Reviewer #1: No

Reviewer #2: No

---

## [Editor Report · Acceptance letter]

31 May 2024

Dear Dr. Liu,

We are delighted to inform you that your manuscript, "Temporal regulation of MDA5 inactivation by Caspase-3 dependent cleavage of 14-3-3η," has been formally accepted for publication in PLOS Pathogens.

Best regards,

Michael Malim

Editor-in-Chief

PLOS Pathogens

orcid.org/0000-0002-7699-2064